# TAMING SELF-TRAINING FOR OPEN-VOCABULARY OBJECT DETECTION

## ABSTRACT

Recent studies have shown promising performance in open-vocabulary object detection (OVD) by utilizing pseudo labels (PLs) from pretrained vision and language models (VLMs). However, teacher-student self-training, a powerful and widely used paradigm to leverage PLs, is rarely explored for OVD. This work identifies two challenges of using self-training in OVD: *noisy PLs from VLMs* and *frequent distribution changes of PLs*. To address these challenges, we propose SAS-Det that tames self-training for OVD from two key perspectives. First, we present a split-and-fusion (SAF) head that splits a standard detection into an open-branch and a closed-branch. This design can reduce noisy supervision from pseudo boxes. Moreover, the two branches learn complementary knowledge from different training data, significantly enhancing performance when fused together. Second, in our view, unlike in closed-set tasks, the PL distributions in OVD are solely determined by the teacher model. We introduce a periodic update strategy to decrease the number of updates to the teacher, thereby decreasing the frequency of changes in PL distributions, which stabilizes the training process. Extensive experiments demonstrate SAS-Det is both efficient and effective. SAS-Det outperforms prior state-of-the-art models of the same scale by a clear margin and achieves 37.4 $AP_{50}$ and 29.1 $AP_r$ on novel categories of the COCO and LVIS benchmarks, respectively.

## 1 INTRODUCTION

Traditional closed-set object detectors (Carion et al., 2020; He et al., 2017; Ren et al., 2015) are restricted to detecting objects with a limited number of categories. Increasing the size of detection vocabularies usually requires heavy human labor to collect annotated data. With the recent advent of strong vision and language models (VLMs) (Li et al., 2021; Radford et al., 2021), open-vocabulary object detection (OVD) (Gu et al., 2022) provides an alternative direction to approach this challenge. Typically, OVD detectors are trained with annotations of base categories and expected to generalize to novel target categories by leveraging the power of pretrained VLMs.

One promising thread of recent studies for OVD (Feng et al., 2022; Gao et al., 2022; Zhao et al., 2022; Wu et al., 2023b) leverages VLMs to obtain pseudo labels (PLs) beyond base categories. But they rarely explore self-training, a powerful and widely used schema for utilizing PLs in closed-set tasks (Tang et al., 2017; Sohn et al., 2020; Xie et al., 2020; Xu et al., 2021). We investigate self-training for OVD and find the vanilla self-training approach does not improve OVD performance due to the following challenges.

First, the typical self-training in closed-set tasks sets a confidence threshold to remove noisy PLs based on the fact that the quality of PLs is positively correlated to their confidences. However, VLMs employed in OVD are pretrained for image-level alignment with texts instead of instance-level object detection that requires the localization ability. Thus, the confidence score from pretrained VLMs is usually not a good indicator for the quality of box locations (i.e., pseudo boxes) provided by PLs. For example, prior studies (Gu et al., 2022; Zhao et al., 2022) show that CLIP (Radford et al., 2021) tends to output imperfect object boxes as predictions with high confidence. Recent methods for OVD (Zhao et al., 2022; Gao et al., 2022) just apply thresholding to VLMs' confidence scores and ignore the poor quality of pseudo boxes, which provides noisy supervision to the model. This issue becomes even worse when self-training is applied directly, since the noise accumulates

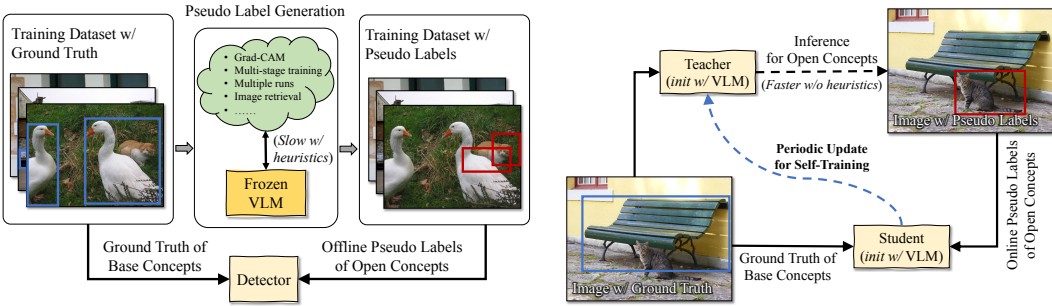

Figure 1: **Left:** Prior PL-based methods for OVD rely on handcrafted heuristics to leverage a frozen VLM for offline pseudo labels. This is usually inefficient and does not allow for improving PLs throughout training. **Right:** We customize self-training and finetune VLMs for OVD, which allows efficient on-the-fly computation of PLs that can be improved throughout training.

which degrades the performance on novel categories. Moreover, these methods handle noisy PLs in the same way as ground truth of base categories during training, which further decreases the performance on base categories (Gao et al., 2022; Lin et al., 2023).

Second, self-training for closed-set object detection (Tang et al., 2017; Xu et al., 2021; Li et al., 2022a) usually follows an online teacher-student manner. In each training iteration, the teacher generates PLs, and the student is trained with a mixture of ground truth and PLs. Then, the teacher is updated by the student with exponential moving average (EMA). However, we find such EMA updates degrade OVD models (see Table 4). Our hypothesis is that, unlike closed-set tasks, OVD provides no ground truth for target categories, and thus, the supervision for target categories is fully decided by the distribution of PLs predicted by the teacher. Hence, the EMA updates change the distribution of PLs in each iteration, making the training unstable.

In this paper, we propose *Self-training And Split-and-fusion head* for open-vocabulary Detection (SAS-Det) to tame self-training for OVD. First, we present a split-and-fusion (SAF) head to handle the noise of PLs. The SAF head splits the standard detection head into two branches: the closed-branch and the open-branch, which are fused at inference. The closed-branch, akin to the standard detection head, comprises a classification module and a box refinement module. It is supervised solely by ground truth from base categories, mitigating the impact of noisy PLs on the performance of base categories. The open-branch is a classification module supervised by class labels of both ground truth and PLs. It acquires complementary knowledge to the closed-branch and can significantly boost the performance when fused with the closed-branch. Moreover, this design circumvents noisy locations of pseudo boxes, reducing the accumulation of noise during self-training.

Second, instead of adopting the vanilla EMA update, we reduce the number of the updates and periodically update the teacher by the student. The quality of our PLs improves along with the periodic updates, and our final PLs are better than those of prior PL-based methods (Gao et al., 2022; Zhao et al., 2022) that introduce external handcrafted steps. Fig. 1 shows the key differences.

The proposed SAS-Det outperforms state-of-the-art OVD models of the same scale by a clear margin on two popular benchmarks, i.e., COCO and LVIS. Without extra handcrafted steps, our pseudo labeling is more efficient than prior methods, i.e., nearly 4 times faster than PB-OVD (Gao et al., 2022) and 3 times faster than VL-PLM (Zhao et al., 2022). Extensive ablation studies demonstrate the effectiveness of the proposed components. On COCO, the thresholding of vanilla self-training decreases the performance of novel categories by 3.6 AP, and the EMA update decreases the performance by 6.9 AP. Instead, SAS-Det eliminates the degradation with two separate detection heads and the periodic update. The fusion of the SAF head boosts the performance by 6.0 AP.

The contributions of this work are summarized as follows. (1) We show two challenges of applying self-training to OVD and propose two simple but effective solutions, i.e., using different detection heads to mitigate the noise in PLs, and using periodic updates to reduce frequency of changes in PLs' distributions. (2) The proposed SAF head for OVD handles the noisy boxes of PLs and enables fusion to improve the performance. (3) We present the state-of-the-art performance on COCO and LVIS under widely used OVD settings and provide detailed analysis of the proposed SAS-Det.

## 2 RELATED WORK

**Vision-language models (VLMs).** VLMs are trained to learn the alignment between images and text in a common embedding space. CLIP (Radford et al., 2021) and ALIGN (Jia et al., 2021) use contrastive losses to learn such alignment on large-scale noisy image-text pairs from the Internet. ALBEF (Li et al., 2021) introduces multi-modal fusion and additional self-supervised objectives. SIMLA (Khan et al., 2022) employs a single stream architecture to achieve multi-level image-text alignment. FDT (Chen et al., 2023) learns shared discrete tokens as the embedding space. These VLMs achieve impressive zero-shot performance on image classification. But due to the gap between the pretraining and detection tasks, VLMs have limited abilities in object detection. In this work, we attempt to close the gap via PLs. There are studies (Kamath et al., 2021a; Li et al., 2022b; Schulter et al., 2023) focusing on aligning any text phrases with objects. But they require visual grounding data that are more expensive than detection annotations. Our work scales up the vocabulary size for object detection without requiring such costly data.

**Open-vocabulary object detection (OVD).** Traditional closed-set object detectors (Redmon & Farhadi, 2017; Ren et al., 2015; He et al., 2017; Carion et al., 2020) achieve great performance but are limited in their vocabulary size. Zero-shot methods (Bansal et al., 2018; Rahman et al., 2020; Zhu et al., 2020a; Yan et al., 2022) increase the vocabulary size but with limited accuracy. Motivated by the strong zero-shot abilities of VLMs, recent efforts focus on OVD. Finetuning-based methods (Zareian et al., 2021; Minderer et al., 2022; Zhong et al., 2022; Kuo et al., 2023; Kim et al., 2023) add detection heads onto pretrained VLMs and then finetune the detector with concepts of base categories. Such methods are simple but may forget the knowledge learned in the pretraining (Arandjelović et al., 2022). Distillation-based methods (Gu et al., 2022; Du et al., 2022; Wu et al., 2023a) introduce additional distillation loss functions that force the output of a detector to be close to that of a VLM and thus avoid forgetting. However, since the distillation losses are not designed for the detection task, they may conflict with detection objectives due to gradient conflicts that are a common issue for multi-task models. Methods like Gao et al. (2022); Zhao et al. (2022); Feng et al. (2022); Wu et al. (2023b) create pseudo labels (PL) of novel concepts as supervision, and do not require extra losses, which sidesteps both catastrophic forgetting and gradient conflicts. But these methods need handcrafted steps to generate high quality PLs, e.g. multiple runs of box regression (Zhao et al., 2022), activation maps (Gao et al., 2022) from Grad-CAM (Selvaraju et al., 2017), image retrieval (Feng et al., 2022), or multi-stage training (Wu et al., 2023b). In this work, we address the two challenges of using self-training for OVD and enable an efficient end-to-end pseudo labeling pipeline. Besides, we point out the noise due to the poor locations of PLs and introduce the SAF head to handle such noise.

**Self-training for object detection.** Weakly-supervised object detection methods (Tang et al., 2017; Wan et al., 2019; Ren et al., 2020) explore online self-training by distilling the knowledge from the model itself. Recent semi-supervised object detection methods (Xu et al., 2021; Liu et al., 2021; Li et al., 2022a) adopt a teacher-student design, where the teacher is an exponential moving average (EMA) of the student. Self-training has been widely explored in the above fields but rarely in OVD. Unlike semi-supervised object detection, OVD encounters two challenges to use self-training, i.e., more noisy PLs and frequent changes in PLs' distributions. This work proposes the SAF head to address the noise, and adopts periodic updates to reduce the frequency of changes in PLs' distribution.

## 3 APPROACH

In open-vocabulary detection, an object detector is trained with bounding boxes and class labels of base categories $\mathcal{C}^B$. At inference, the detector is used for detecting objects of open concepts including $\mathcal{C}^B$ and novel categories $\mathcal{C}^N$, where $\mathcal{C}^B \cap \mathcal{C}^N = \varnothing$. To make such detection practical, recent studies (Zareian et al., 2021; Gu et al., 2022; Feng et al., 2022; Lin et al., 2023) adopt extra data (e.g., image-text pairs and image-level tags) and/or external VLMs. We follow their settings and leverage the pretrained CLIP (Radford et al., 2021) to build our OVD detector.

### 3.1 ADAPTING CLIP TO OVD

In this section, we introduce how to adapt the ResNet-based CLIP into a Faster-RCNN (Ren et al., 2015) (C4) detector. For simplicity, we do not use FPN (Lin et al., 2017), but it can be incorporated

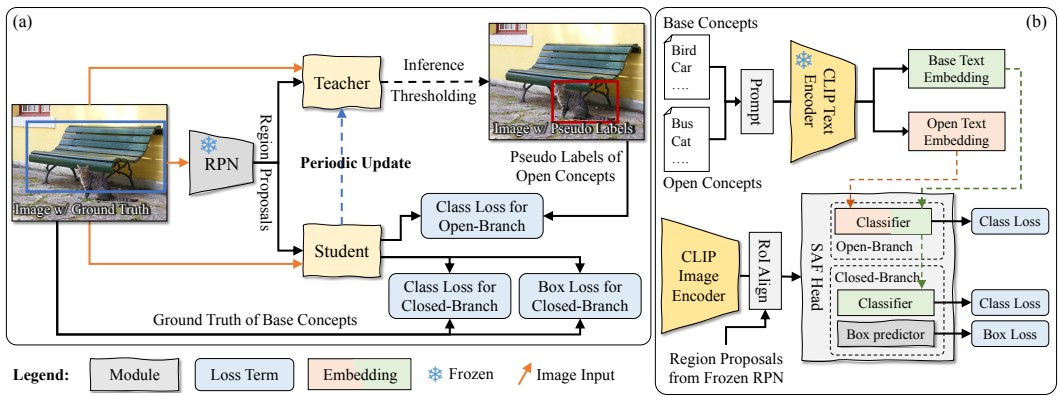

Figure 2: **(a) Pipeline of our self-training.** The teacher and the student are models of the same architecture. They are initialized with the same pretrained CLIP model. The teacher generates PLs that are used to train the student, and the student updates the teacher periodically. **(b) Structure of our detector.** The proposed SAF head is put on top of a CLIP image encoder. The open- and closed-branches take the text embeddings from a CLIP text encoder as classifier.

with learnable gating from Flamingo Alayrac et al. (2022). Similarly, ViT-based CLIP models can be adapted to detectors following (Li et al., 2022c).

**Region proposals from an external RPN.** CLIP is pretrained via image-text alignment. As shown in Sect. 4.3, finetuning pretrained backbones to get region proposals decreases the performance, probably because such finetuning breaks the image-text alignment learned in the pretraining. To address this problem, F-VLM (Kuo et al., 2023) freezes the pretrained backbone and only finetunes detection heads. But this solution limits the capacity of the detector. Unlike F-VLM, we follow the approach of Singh et al. (2018); Zhong et al. (2022) and employ an external RPN to generate region proposals, which is trained with ground truth boxes of base categories. Prior studies adopt RPN mainly to accelerate inference (Singh et al., 2018) or to improve region recognition (Zhong et al., 2022). By contrast, we aim to leverage the RPN to preserve the knowledge learned in the pretraining for better self-training.

**Text embeddings as the classifier.** For the $i$-th region proposal from the external RPN, we apply RoIAlign (He et al., 2017) on the 4th feature maps of CLIP's ResNet to get the proposal features. Then, the features are fed to the last ResNet block and the attention pooling of CLIP to get the region embedding $r_i$, which is later used for classification. Following prior studies (Gu et al., 2022; Zhong et al., 2022; Lin et al., 2023), we convert a set of given concepts with prompt engineering into CLIP text embeddings, which act as classifiers, as shown in Fig. 2b. A fixed all-zero embedding is adopted for the background category. Assuming $t_c$ is the embedding of the $c$-th category, the probability of $r_i$ to be classified as the $c$-th category is,

$$p_{i,c} = \frac{\exp(\langle r_i, t_c \rangle / \tau)}{\sum_{j=0}^{C-1} \exp(\langle r_i, t_j \rangle / \tau)}, \tag{1}$$

where $\langle \cdot, \cdot \rangle$ denotes the cosine distance, $C$ denotes the vocabulary size, and $\tau$ denotes the temperature. In our detector, $r_i$ may be further fed to a box refinement module to predict the box shift based on the region proposal. With the above adaptions, our initial detector gains zero-shot detection ability to some extent. Please refer to Appendix A.2 for quantitative evaluations.

## 3.2 TAMING SELF-TRAINING

Although self-training has been widely explored in closed-set object detection, using it for OVD presents two challenges, i.e., noisy PLs and frequent changes of PLs' distributions. In the following, we describe our self-training pipeline and how to address the two challenges.

**Self-training pipeline.** Fig. 2a illustrates the pipeline of our self-training with a teacher-student manner. Training images are first fed into the RPN to obtain region proposals. Then, the teacher

model runs inference on those proposals, where the resulting predictions with confidences above a threshold are selected as PLs. The student model adopts the same region proposals, and is supervised with both base ground truth and PLs generated by the teacher. The teacher is periodically updated with the parameters of the student model.

**Handling noise in PLs.** The noise of PLs from VLMs exists in both classification and localization. It is a common practice to filter PLs with classification confidence, but this only addresses noise in classification. To reduce the noise in locations of PLs, we exclude the noisy boxes of PLs from the training. That is, the classification loss is calculated on class labels of both ground truth and PLs, but the box regression loss is calculated only on ground truth boxes. Besides, to mitigate the impact of PLs on the performance of base categories, we propose SAF head and elaborate it in Sect. 3.3.

**Reducing changes in PLs' distributions with periodic updates.** The teacher model is updated with the student to enable self-training. The exponential moving average (EMA) is a widely used approach for object detection (Li et al., 2022a; Liu et al., 2021; Xu et al., 2021), which updates the teacher in every iteration. But we observed empirically that it does not benefit OVD (Table 4). We hypothesize that, unlike semi-supervised object detection, OVD has no ground truth for the target categories. Thus, the distribution of the target data (i.e., PLs) is fully determined by the teacher. The EMA update changes the PLs' distribution in each iteration, resulting in unstable training. As a solution, we periodically update the teacher after a set number of iterations to maintain consistent distributions for PLs between updates. We call this strategy as the periodic update and show it outperforms the EMA update by a large margin in Sect. 4.3.

## 3.3 SPLIT-AND-FUSION (SAF) HEAD

Our SAF head first splits a detection head into two branches, i.e. "closed-branch" and "open-branch", with the goal to better handle noisy PLs during training. At inference, predictions from both heads are fused to boost the performance.

**Splitting the detection head.** The closed-branch follows a standard detection head with a classification module and a class-agnostic box refinement module. The former module classifies region proposals based on Eq. 1, and the latter one refines the proposal boxes for better locations. We train the closed-branch with boxes and class labels of ground truth for $\mathcal{C}^B$ using standard detection losses, which include a cross entropy loss for classification and a box regression loss for localization. Since no PLs are used to train the closed-branch, the noise of PLs is unlike to impact its performance on $\mathcal{C}^B$. Moreover, as shown in Gu et al. (2022); Zareian et al. (2021), box regression modules trained on $\mathcal{C}^B$ can generalize to novel categories that are unseen during training. Therefore, the closed-branch is able to provide generalized boxes, as well.

The open-branch only contains a classification module. It is trained using only the cross-entropy loss with class labels from both $\mathcal{C}^B$ and PLs, hence, learning broader concepts beyond $\mathcal{C}^B$. Unlike distillation losses (Gu et al., 2022; Du et al., 2022), all losses for the two branches are originally designed for detection and are unlikely to conflict with each other. When generating PLs, we use the classification scores from the open-branch and bounding boxes from the closed-branch. Boxes of PLs are not directly used in our losses but are involved to select foreground proposal candidates for the classification loss.

**Fusing complementary predictions.** The open- and closed-branches are trained in different ways and learn complementary knowledge. Therefore, we fuse their predictions with the geometric mean at inference time. Specifically, assuming $p_{i,c}^{\mathrm{open}}$ and $p_{i,c}^{\mathrm{closed}}$ are prediction scores of the open- and closed-branches, respectively, the final score is calculated as

$$p_{i,c}^{\mathrm{fused}} = \begin{cases} (p_{i,c}^{\mathrm{closed}})^{(1-\alpha)} \cdot (p_{i,c}^{\mathrm{open}})^{\alpha}, & \text{if } i \in \mathcal{C}^B \\ (p_{i,c}^{\mathrm{closed}})^{\alpha} \cdot (p_{i,c}^{\mathrm{open}})^{(1-\alpha)}, & \text{if } i \in \mathcal{C}^N \end{cases} \tag{2}$$

where $\alpha \in [0, 1]$ balances the two branches. The indices $i, c$ are the same as in Eq. 1. We keep a list of known base categories $\mathcal{C}^B$ and take other categories beyond the list as novel. It's important to note that fusion is a common strategy. For instance, F-VLM (Kuo et al., 2023) employs score fusion to enhance the location quality of CLIP's predictions. In contrast, our approach focuses on fusing the complementary predictions from the two branches. The primary contribution of our method lies in learning what to fuse rather than the fusion process itself.

Table 1: Comparison with state-of-the-art methods on COCO-OVD. We group methods into two blocks. The first block contains methods using stronger backbones or detector architectures than ours. The second block contains models of the same scale as ours. Training setup indicates training iterations (N×) and data augmentations. Large Scale Jittering (LSJ) (Ghiasi et al., 2021) is a stronger data augmentation than Detectron2's default.

| Method | Training Setup | Backbone | Detector | $\text{AP}_{50}^{novel}$ | $\text{AP}_{50}^{base}$ | $\text{AP}_{50}^{all}$ |
|---|---|---|---|---|---|---|
| ViLD (Gu et al., 2022) | 16×+LSJ | RN50-FPN | FasterRCNN | 27.6 | 59.5 | 51.2 |
| VL-PLM (Zhao et al., 2022) | 1×+Default | RN50-FPN | FasterRCNN | 32.3 | 54.0 | 48.3 |
| F-VLM (Kuo et al., 2023) | 0.5×+LSJ | RN50-FPN | FasterRCNN | 28.0 | - | 39.6 |
| OV-DETR (Zang et al., 2022) | (Not Given) | RN50 | Deform. DETR | 29.4 | 61.0 | 52.7 |
| CORA (Wu et al., 2023b) | 3×+Default | RN50 | DAB-DETR | **35.1** | 35.5 | 35.4 |
| OVR-CNN (Zareian et al., 2021) | (Not Given) | RN50-C4 | FasterRCNN | 22.8 | 46.0 | 39.9 |
| RegionCLIP (Zhong et al., 2022) | 1×+Default | RN50-C4 | FasterRCNN | 26.8 | 54.8 | 47.5 |
| Detic (Zhou et al., 2022) | 1×+Default | RN50-C4 | FasterRCNN | 27.8 | 51.1 | 45.0 |
| PB-OVD (Gao et al., 2022) | 6×+Default | RN50-C4 | FasterRCNN | 30.8 | 46.1 | 42.1 |
| VLDet (Lin et al., 2023) | 1×+LSJ | RN50-C4 | FasterRCNN | 32.0 | 50.6 | 45.8 |
| BARON (Wu et al., 2023a) | 1×+Default | RN50-C4 | FasterRCNN | 33.1 | 54.8 | 49.1 |
| SAS-Det (Ours) | 1×+Default | RN50-C4 | FasterRCNN | **37.4** | 58.5 | 53.0 |

Table 2: Comparison with state-of-the-art methods on LVIS-OVD. We group methods based on the scale of backbones. Training setup contains training iterations (N×) and data augmentations. LSJ (Ghiasi et al., 2021) is a stronger data augmentation than Detectron2's default.

| Method | Training Setup | Backbone | Detector | $\text{AP}_r$ | $\text{AP}_c$ | $\text{AP}_f$ | AP |
|---|---|---|---|---|---|---|---|
| ViLD (Gu et al., 2022) | 16×+LSJ | RN50-FPN | FasterRCNN | 16.7 | 26.5 | 34.2 | 27.8 |
| DetPro (Du et al., 2022) | 2×+Default | RN50-FPN | FasterRCNN | 20.8 | 27.8 | 32.4 | 28.4 |
| F-VLM (Kuo et al., 2023) | 9×+LSJ | RN50-FPN | FasterRCNN | 18.6 | - | - | 24.2 |
| BARON (Wu et al., 2023a) | 2×+Default | RN50-C4 | FasterRCNN | 17.3 | 25.6 | 31.0 | 26.3 |
| SAS-Det (Ours) | 2×+Default | RN50-C4 | FasterRCNN | **20.9** | 26.1 | 31.6 | 27.4 |
| ViLD (Gu et al., 2022) | 16×+LSJ | RN152-FPN | FasterRCNN | 19.8 | 27.1 | 34.5 | 28.7 |
| OWL-ViT (Minderer et al., 2022) | 12×+LSJ | ViT-L/14 | OWL-ViT | 25.6 | - | - | 34.7 |
| F-VLM (Kuo et al., 2023) | 9×+LSJ | RN50x4-FPN | FasterRCNN | 26.3 | - | - | 28.5 |
| CORA (Wu et al., 2023b) | 3×+Default | RN50x4 | DAB-DETR | 22.2 | - | - | - |
| SAS-Det (Ours) | 2×+Default | RN50x4-C4 | FasterRCNN | **29.1** | 32.4 | 36.8 | 33.5 |

# 4 EXPERIMENTS

## 4.1 EXPERIMENT SETUP

**Datasets.** We conduct experiments on the two popular OVD benchmarks based on COCO (Lin et al., 2014) and LVIS (Gupta et al., 2019). For the COCO dataset, 65 out of 80 categories are divided into 48 base categories and 17 novel categories. We report results on the validation set. This setting is denoted as COCO-OVD. For the LVIS dataset, 337 rare categories are used as the novel concepts and the remaining 866 categories (frequent and common) as the base concepts. We report results on the whole LVIS validation set. This setting is denoted as LVIS-OVD.

**Evaluation metrics.** Following prior studies (Zareian et al., 2021; Gu et al., 2022; Zhong et al., 2022; Wu et al., 2023a), we report AP at IoU threshold of 0.5 for COCO-OVD. $\text{AP}_{50}^{novel}$, $\text{AP}_{50}^{base}$ and $\text{AP}_{50}^{all}$ are the metrics for the novel, the base and all (novel + base) categories, respectively. For LVIS-OVD, we report the mean AP averaged on IoUs from 0.5 to 0.95. $\text{AP}_r$, $\text{AP}_c$, $\text{AP}_f$ and AP are the metrics for rare, common, frequent, and all categories.

**Implementation details.** We built our detector on Faster RCNN with the CLIP version of ResNet (RN50-C4 or RN50x4-C4) as the backbone. Following prior studies (Zang et al., 2022; Zhao et al., 2022; Zhou et al., 2022), we assumed the novel concepts were available during training and took them as our open concepts, but no annotations of novel concepts are used. Our method was imple-

mented on Detectron2 (Wu et al., 2019) and trained with 8 NVIDIA A6000 GPUs. We trained our models with the $1\times$ schedule (90k iterations) for COCO-OVD and the $2\times$ schedule for LVIS-OVD. The batch size was 16 with an initial learning rate of 0.002. The default data augmentation of Detectron2 was applied. Loss terms were equally weighted. By default, the teacher models were updated three times, usually together with the decreases of the learning rate.

## 4.2 COMPARISON WITH THE EXISTING METHODS

**COCO-OVD.** We compare our method with prior work on COCO in Table 1. With the same backbone and detector, SAS-Det outperforms the prior state-of-the-art BARON (Wu et al., 2023a) by 4.3 $AP_{50}^{novel}$ on novel categories and 3.7 $AP_{50}^{base}$ on base categories. Probably, BARON distills knowledge from VLMs, but distillation may lead to gradient conflicts and degrade the performance (Zhao et al., 2022). Compared to VLDet (Lin et al., 2023), SAS-Det gains an improvement of 7.9 $AP_{50}^{base}$. It is likely that VLDet employs noisy pseudo labels to train the detection head, impacting the detection of base classes in both classification and localization. In contrast, the two-branch design of our SAF head allows to train the closed-branch with just ground-truth of base classes and reduces the impact of the noise from pseudo labels. For more analysis on where our improvements come, please refer to Sect. 4.3 and Table 3. The first block of Table 1 provides results for methods with stronger backbones or detector architectures. When compared with those methods, SAS-Det still achieves the leading performance. Those results clearly demonstrate the effectiveness of SAS-Det.

**LVIS-OVD.** We provide the main results on LVIS in Table 2. When using ResNet50 as the backbone, SAS-Det achieves similar performance as the state-of-the-art DetPro (Du et al., 2022). DetPro proposes learnable prompts to generate better text embeddings as the classifier for OVD. Our method is orthogonal and can leverage DetPro's prompts for further improvement. The first block of Table 2 also shows that SAS-Det outperforms ViLD by a large margin on novel categories (indicated by $AP_r$) but gets lower $AP_f$ for base categories. This is also observed in the recent method BARON (Wu et al., 2023a). The performance gap is probably due to the training setup. ViLD is trained for 8 times more iterations and adopts a strong data augmentation method (Ghiasi et al., 2021), both of which benefit detection on base categories. In contrast, BARON and our approach adopt shorter training and standard data augmentations. When replacing the pretrained CLIP ResNet50 with ResNet50x4 (Radford et al., 2021) as the backbone, we improve $AP_r$ by 8.2 from 20.9 to 29.1, which demonstrates SAS-Det scales up nicely with stronger pretrained VLM backbones.

## 4.3 ABLATION STUDIES

In this section, the default *baseline* shares the same architecture and the training as our SAS-Det (with RN50-C4 as the backbone), except that the SAF head is replaced with a single detection head. The localization module of the detection head is trained with base ground truth boxes only, but the classification module is trained with class labels of both ground truth and PLs. Thus, the *baseline* provides a naive solution to exclude noisy pseudo boxes from the training. All evaluations are conducted on COCO.

**External RPN.** We leverage an external RPN to generate region proposals so that finetuning focuses on region-text alignment that is similar to the pretraining task. In this way, the detector is unlikely to forget the knowledge obtained in the pretraining. To validate the effectiveness of the external RPN, we follow F-VLM (Kuo et al., 2023) to train a detector without an external RPN (See Appendix A.1 for more details). Table 3 compares *baseline* with the detector in the row of *(1)*. As shown, without the external RPN, the performance drops on both novel and base categories. External RPN requires additional computational costs. In Appendix B.1, we provide a solution to remove the cost while keeping the performance.

**Removing boxes of PLs from training.** Our *baseline* handles location noise of PLs' boxes by directly excluding them from the training. As shown in Table 3, *baseline* outperforms *(2)*'s detector. The only difference between them is the removal of pseudo boxes from training. This result clearly shows that simply removing the pseudo boxes is an effective way to deal with location noise.

**Splitting the detection head.** As shown in Table 3, compared to *baseline*, the open-branch of the proposed SAF head in *(5)* achieves similar performance on novel categories. This indicates that the

Table 3: Ablation studies to analyze the effect of each component of SAS-Det on COCO-OVD. Results *(4)*, *(5)*, and *(6)* denote different outputs from the same model.

| Ablation | $\text{AP}_{50}^{novel}$ | $\text{AP}_{50}^{base}$ | $\text{AP}_{50}^{all}$ |
|---|---|---|---|
| *Baseline* | 31.4 | 55.7 | 49.4 |
| *(1)* No external RPN, train the backbone for region proposals | (-6.0) 25.4 | 53.4 | 46.1 |
| *(2)* Noisy boxes of PLs as supervision for box regression | (-3.6) 27.8 | 55.2 | 48.0 |
| *(3)* No PLs, train with base data only | (-8.2) 23.2 | 56.9 | 48.1 |
| *(4)* W/ SAF head, use closed-branch's predictions ($p_{i,c}^{\text{closed}}$ in Eq. 2) | (-5.8) 25.6 | 57.9 | 49.5 |
| *(5)* W/ SAF head, use open-branch's predictions ($p_{i,c}^{\text{open}}$ in Eq. 2) | (+0.5) 31.9 | 57.4 | 50.7 |
| *(6)* W/ SAF head, use fused predictions ($p_{i,c}^{\text{fused}}$ in Eq. 2) | (+6.0) 37.4 | 58.5 | 53.0 |

Table 4: Comparison of different strategies for updating the teacher in self-training on COCO-OVD.

| Update strategy | $\text{AP}_{50}^{novel}$ | $\text{AP}_{50}^{base}$ | $\text{AP}_{50}^{all}$ |
|---|---|---|---|
| *Baseline* (w/ periodic update) | 31.4 | 55.7 | 49.4 |
| *(7)* No teacher and no pseudo labels | (-8.2) 23.2 | 56.9 | 48.1 |
| *(8)* Not update the teacher | (-1.8) 29.6 | 55.9 | 49.1 |
| *(9)* Take the EMA of the student as the teacher | (-6.9) 24.5 | 53.9 | 46.2 |
| *(10)* Replace the teacher with the student every iteration | (-23.4) 8.0 | 53.3 | 41.5 |

split handles the noise in PLs' location as well as the naive solution of *baseline*. This is plausible because both solutions exclude the noisy boxes of PLs from training.

Additionally, the open-branch gains 1.7 $\text{AP}_{50}^{base}$ on base categories. Compared to *(3)*'s, the closed-branch in *(4)* gain improvements in terms of both $\text{AP}_{50}^{novel}$ and $\text{AP}_{50}^{base}$. Note that *(4)*'s and *(3)*'s heads are trained with the same data. Based on those results, the split benefits OVD beyond handling the location noise. One advantage is that, due to the split, the closed-branch is less likely to be influenced by the noise of PLs and learn how to better localize objects.

**Fusing predictions.** Our SAF head fuses predictions of the open-branch and the closed-branch at inference time. As shown in Table 3, *(6)*'s outperforms all the others by a large margin with the prediction fusion. Note that *(4)*'s, *(5)*'s and *(6)*'s refer to the different outputs of the same model. The improvement can be attributed to the fact that the two branches are trained on different sets of data and learn complementary knowledge. The closed-branch is trained with class labels and boxes of base categories. The open-branch is trained with class labels of more vocabularies including base and novel classes. Thus, the former learns more about how to localize objects and how to detect base classes. The latter learns more about how to detect new objects, which complements the former.

**Update strategies for the teacher model.** The teacher is updated with the student model to improve PLs during finetuning. We evaluated several strategies for updating the teacher model in Table 4 and summarize our findings as follows. First, the EMA update in *(9)*, which is shown effective and widely used in semi-supervised object detection (Xu et al., 2021; Liu et al., 2021; Li et al., 2022a), is just slightly better than training without PLs in *(7)*. It is worse than no update to the teacher (i.e. no self-training) in *(8)*. Second, if we replace the teacher with the student every iteration as *(10)*, $\text{AP}_{50}^{novel}$ drops to 8.0 that is much lower than that number in *(7)*. These findings indicate that it's harmful to change the teacher frequently. We hypothesize that frequent updates change the distribution of PLs too often and makes the training unstable. Last but not least, by reducing the number of updates, our periodic update significantly outperforms the EMA update in OVD. Appendix A.3 shows how different numbers of updates affect the performance.

### 4.4 FURTHER ANALYSIS

**Evaluation on the retained 15 COCO categories.** Following prior studies (Zang et al., 2022; Zhao et al., 2022; Zhou et al., 2022), SAS-Det assumes novel concepts are available during training and

Table 5: Generalization capability of OVD methods on retained 15 COCO categories. Text embeddings of 80 COCO categories are used as the classifier. Numbers are copied from MEDet (Chen et al., 2022).

| Method | $AP_{50}^{retain}$ | $AP_{50}^{novel}$ | $AP_{50}^{coco}$ |
|---|---|---|---|
| OVR-CNN Zareian et al. (2021) | 11.5 | 22.9 | 38.1 |
| Detic-80 Zhou et al. (2022) | 11.5 | 27.3 | 38.3 |
| MEDet Chen et al. (2022) | 18.6 | 32.6 | 42.4 |
| Ours | **23.0** | **37.1** | **47.2** |

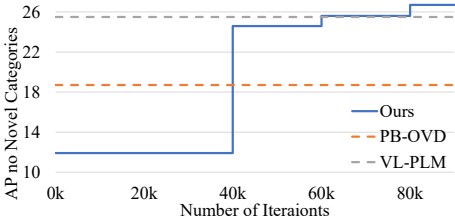

Figure 3: Quality of PLs during training.

Table 6: Mean time cost to get PLs per image. The quality of PLs on the validation set of COCO-OVD are provided for reference. We report the quality of our PLs after the last update.

| Method | Time (s) | $AP_{50}^{novel}$ |
|---|---|---|
| PB-OVD Gao et al. (2022) | 0.4848 | 18.7 |
| VL-PLM Zhao et al. (2022) | 0.4456 | 25.5 |
| Ours | **0.1308** | **26.7** |

achieves good performance. A natural question is if SAS-Det generalizes to alien concepts that are unknown before the evaluation. To answer the question, we follow the evaluation protocol of MEDet (Chen et al., 2022) where an OVD detector is evaluated on the whole COCO validation set with all 80 COCO concepts. $AP_{50}^{retain}$, $AP_{50}^{novel}$ and $AP_{50}^{coco}$ are reported as APs averaged on 15 retrained, 17 novel and all 80 COCO categories, respectively. As shown in Table 5, SAS-Det achieves leading performance by a clear margin, which indicates the good generalization capability of the proposed method.

**PL's quality during self-training.** We compare our PLs and the prior state-of-the-art PLs from VL-PLM (Zhao et al., 2022) on the COCO validation set in terms of PLs' quality. Following VL-PLM, we take $AP_{50}^{novel}$ as the metric and evaluate our teacher models during training. As illustrated in Fig. 3, the quality of our initial PLs is not as good as VL-PLM's. But ours get close to VL-PLM's after two updates and become the better by the third update. This is because the teacher is initialized with CLIP, which is not pretrained for detection, and thus cannot provide high quality PLs. With updates, the teacher is equipped with the knowledge about detection and generates promising PLs.

**Time cost of pseudo labeling.** We run two major PL-based methods (Gao et al., 2022; Zhao et al., 2022) on our computational environment and compare them with our pseudo labeling in terms of the time cost in Table 6. As shown, our method is 3× faster than VL-PLM (Zhao et al., 2022) and nearly 4× faster than PB-OVD (Gao et al., 2022), offering better PLs.

## 5 CONCLUSION

In this paper, we highlight two challenges associated with applying self-training to OVD: 1) noisy PLs from pretrained VLMs and 2) frequent changes of PLs' distributions. To address the challenges, we introduce a split-and-fusion (SAF) head and implement periodic updates. The SAF head splits a standard detection head into an open-branch and a closed-branch. The open-branch is trained with the class labels of ground truth and PLs. The closed-branch is trained with both class labels and boxes of ground truth. This approach effectively eliminates the location noise of PLs during training. Furthermore, the SAF head acquires complementary knowledge from different sets of training data, enabling fusion to enhance performance. The periodic updates reduce the number of updates to the teacher models, thereby decreasing the frequency of changes in PLs' distributions. We demonstrate that the proposed method SAS-Det is both efficient and effective. Our pseudo labeling is much faster than prior PL-based methods (Gao et al., 2022; Zhao et al., 2022). SAS-Det also outperforms state-of-the-art models of the same scale on both COCO and LVIS for OVD.

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

APPENDIX

In this appendix, we supplement the main paper as follows.

- Sect. A.1 elaborates how to train a detector for open-vocabulary object detection (OVD) without an external RPN, which is used in "**External RPN**" of Sect. 4.3 in the main paper.

- Sect. A.2 shows that the quality of pseudo labels (PLs) from the initial teacher and how to improve initial PLs for Sect. 3.1.

- Sect. A.3 elaborates the exact iterations when we update the teacher model for "**Update strategies for the teacher model**" of Sect. 4.3.

- Sect. A.4 shows the performance of our method with different initial weights.

- Sect. B.1 shows how to remove the extra computations cost of the extra RPN.

- Sect. B.2 compares the proposed SAS-Det and visual grounding methods.

- Sect. B.3 shows how well SAS-Det preserves knowledge learned in the pretraining.

- Sect. B.4 provides extra ablation studies on LVIS.

- Sect. B.5 explores stronger external RPNs for SAS-Det.

- Sect. B.6 explores to use external PLs at the beginning of self-training

- Sect. C discusses some limitations of this work and potential solutions.

- Sect. D.1 provides qualitative results of PLs for good and failure cases.

- Sect. D.2 provides qualitative results of our OVD detector for good and failure cases.

## A    EXTRA DETAILS FOR THE MAIN PAPER

### A.1    OUR DETECTOR FOR OVD WITHOUT AN EXTERNAL RPN

This section explains how we train a detector without an external RPN. This approach is compared with the *baseline* in paragraph "**External RPN**" of Sect. 4.3. As shown in Fig. 4, we divide the training into two stages: (a) In the first stage, we put a RPN box head on top of a pretrained but frozen CLIP image encoder and only train the box head with a box loss. The RPN box head outputs region proposals with objectness scores. The box loss consists of a foreground classification term and a box regression term. (b) In the second stage, the RPN box head and a detection head are built on the same CLIP image encoder. No modules are frozen. The whole model is trained the same way as *baseline* in the main paper with online pseudo labels from a teacher model.

Table 7 provides the performance of the first and the second training stages. We evaluate the model of the 1st stage by directly classifying region proposals with text embeddings. The model of the 1st stage can be regarded as the initial teacher of the 2nd stage. It achieves similar performance as the initial teacher of *baseline*, which indicates that initial PLs of *baseline* and the 2nd stage share similar qualities. However, *baseline* outperforms the 2nd stage's model on both base and novel categories. Such results clearly demonstrate that an external RPN is important to our detector and training.

Table 7: Performance of OVD detectors without an external RPN.

| Models | $AP_{50}^{novel}$ | $AP_{50}^{base}$ | $AP_{50}^{all}$ |
|---|---|---|---|
| First stage (no external RPN) | 10.5 | 12.7 | 12.1 |
| Second stage (no external RPN) | 25.4 | 53.4 | 46.1 |
| Initial teacher of *baseline* (w/ an external RPN) | 11.9 | - | - |
| *baseline* (w/ an external RPN) | **31.4** | 55.7 | 49.4 |

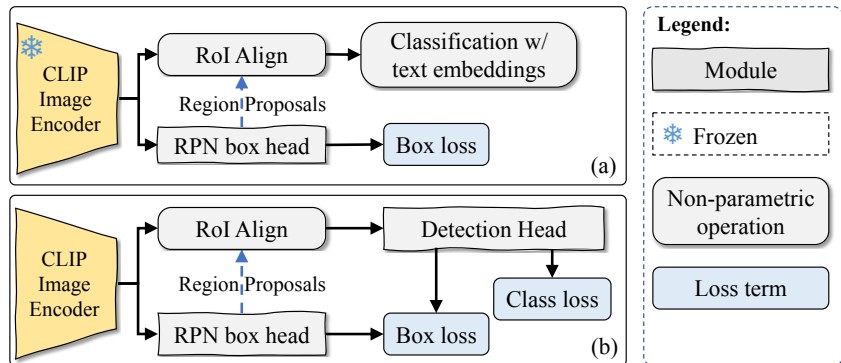

Figure 4: Two stage training for an OVD detector without an external RPN. **(a)** In the first stage, only RPN box head is trained. Text embeddings are used for classification at inference time. **(b)** In the second stage, no modules are frozen.

Table 8: The quality of PLs generated by the initial model w/ or w/o the RPN fusion on novel categories of the COCO and LVIS datasets.

| Method | $AP_{50}^{novel}$ (COCO) | $AP_r$ (LVIS) |
|---|---|---|
| w/o RPN scores | 3.8 | 1.9 |
| w/ RPN scores | **11.9** | **9.1** |

## A.2 Improving initial pseudo labels with RPN scores

In the self-training process, we initialize the teacher model with pretrained CLIP weights and generate PLs based on Eq. 1. However, the initial teacher cannot provide high quality PLs because CLIP is weak at localizing objects and has poor zero-shot detection ability (Gu et al., 2022; Zhao et al., 2022). Similar to VL-PLM (Zhao et al., 2022), we average the objectness scores from the external RPN with the prediction scores from the teacher model. Assuming $s_i^{\text{RPN}}$ is the objectness score of the $i$-th region proposal, the averaged prediction score is $\hat{p}_{i,c} = (s_i^{\text{RPN}} + p_{i,c})/2$ where $c$ refers to the $c$-th category. Table 8 provides the quantitative results for whether or not to use the RPN fusion. As shown, without the fusion, the quality of PLs significantly drops.

## A.3 When to update the teacher model.

In this section, we discuss the timing of updates to the teacher model during training. We trained our detectors with different times of updates to teacher models and provides the exact iterations when we update the teacher model for COCO-OVD in Table 9. Generally, we consider the learning rate schedule and distribute updates as evenly as possible during training. As shown in Table 9, too many updates, e.g., 8 or 4 updates, lead to performance drops mainly due to the following. First, similar as the aforementioned EMA update, too many updates change the distribution of PLs too often and make the training unstable. Second, the more updates, the earlier an update happens. However, the student model is not well trained at the early stage of the training and thus is not good enough to update the teacher. Table 9 shows that 2 and 3 updates achieve similar performance. But we set 3 updates as default to include as many updates as possible, considering that the only overhead of our update is to copy the weights from the student to the teacher. For LVIS-OVD, we find that a later update helps and only conduct the update when the learning rate changes. For the $2\times$ training setup, the teacher is updated at 120k and 160k iterations.

## A.4 Initial weights.

We follow RegionCLIP (Zhong et al., 2022) and adopt its improved CLIP weights to initialize methodAbbr. However, we noticed that RegionCLIP's pretraining includes LVIS base boxes

Table 9: Iterations to update the teacher model for different number of updates on COCO-OVD. The updates are usually conducted at 6k and 8k iterations when the learning rate decreases.

| # Updates | Iterations to update | $AP_{50}^{novel}$ |
|---|---|---|
| 8 | Every 1k iterations | 27.6 |
| 4 | Every 2k iterations | 30.6 |
| 3 (*baseline*) | 4k,6k,8k | 31.4 |
| 2 | 4k,8k | **31.6** |
| 1 | 5k | 30.9 |
| 0 (No update) | N/A | 29.6 |

Table 10: Performance of *baseline* using different pretrained weights as initialization.

| Initial weights | COCO-OVD | | | LVIS-OVD | | | |
|---|---|---|---|---|---|---|---|
| | $AP_{50}^{novel}$ | $AP_{50}^{base}$ | $AP_{50}^{all}$ | $AP_r$ | $AP_c$ | $AP_f$ | AP |
| From original CLIP | 31.1 | 53.5 | 47.7 | 19.6 | 23.6 | 30.6 | 25.6 |
| From RegionCLIP pretraining | **31.4** | 55.7 | 49.4 | **19.8** | 25.3 | 31.5 | 26.8 |

that may include boxes for novel categories of COCO. To avoid potential data leakage, for our experiments on COCO-OVD, we followed RegionCLIP's pretraining procedure and improve a CLIP model with COCO base boxes only. Besides, On COCO-OVD, we report results with Soft NMS (Bodla et al., 2017) as RegionCLIP (Zhong et al., 2022) did.

Table 10 compares original CLIP weights with RegionCLIP version of CLIP weights as the initialization of *baseline*. As shown, initialized with either of them, the detectors achieve similar performance on novel categories on both the COCO and LVIS datasets. This is probably because our finetuning with PLs closes the gap between CLIP's and RegionCLIP's pretraining. Since RegionCLIP does not benefit our method, it is still fair to compare SAS-Det with other methods that uses the original CLIP. Table 10 also shows that initialization with RegionCLIP improves the performance on base categories. Since RegionCLIP adopts the boxes of base categories in its pretraining, it actually provides longer training on base categories. We attribute the improvement on base categories to the longer training.

## B ADDITIONAL EXPERIMENTS

### B.1 REMOVING EXTRA COMPUTATION COSTS

Our method leverages an external RPN that requires extra computation costs. But we can alleviate this cost by training a new detector with our best pseudo labels. To demonstrate this solution, we first adopted trained SAS-Det to generate PLs on COCO. Then, using the PLs, we trained a Faster RCNN with ResNet50 as the backbone without the external RPN (denoted as "Faster RCNN + PLs"). As shown in Table 11, the model achieves similar performance as SAS-Det and avoids extra computation costs. Note that the above solution can be applied to any detectors, e.g. DETR and its variants, as well as Faster RCNN.

### B.2 COMPARISON WITH VISUAL GROUNDING MODELS

There are some recent studies (Kamath et al., 2021b; Li et al., 2022b; Cai et al., 2022) focusing on large-scale pretraining for visual grounding where text phrases in a whole caption are aligned with objects. They usually train their models with multiple datasets, including detection data, visual grounding data (Li et al., 2022b), and image-text pairs. Visual grounding data requires the association between each box and each specific text phrase, which is much more expensive than detection annotations. The datasets GoldG+ and GoldG, introduced in (Li et al., 2022b),are two widely used

Table 11: Training Faster RCNN with PLs from SAS-Det on COCO-OVD

| Method | $\text{AP}_{50}^{novel}$ | $\text{AP}_{50}^{base}$ | $\text{AP}_{50}^{all}$ |
|---|---|---|---|
| SAS-Det (w/ exteranl RPN) | 37.4 | 58.5 | 53.0 |
| Faster RCNN + PLs (w/o exteranl RPN) | 38.1 | 58.3 | 53.0 |

Table 12: Comparison with visual grounding methods on LVIS minival. Our method adopts pretrained CLIP model that is supervised with image-text pairs automatically collected from the Internet. Reference for methods: MDETR (Kamath et al., 2021b), GLIP (Li et al., 2022b), X-DETR (Cai et al., 2022)

| Method | Training Data | Backbone | Detector | $\text{AP}_r$ | $\text{AP}_c$ | $\text{AP}_f$ | AP |
|---|---|---|---|---|---|---|---|
| MDETR | LVIS, GoldG+ | RN101 | DETR (Carion et al., 2020) | 7.4 | 22.7 | 25.0 | 22.5 |
| GLIP | O365, GoldG, Cap4M | Swin-T | DyHead (Dai et al., 2021) | 20.8 | 21.4 | 31.0 | 26.0 |
| X-DETR | LVIS, GoldG+, CC, LocNar | RN101 | Def.DETR (Zhu et al., 2020b) | 24.7 | 34.6 | 35.1 | 34.0 |
| Ours | LVIS Base | RN50-C4 | FasterRCNN | 20.9 | 26.1 | 31.6 | 27.4 |
| Ours | LVIS Base | RN50x4-C4 | FasterRCNN | **27.3** | 30.7 | 35.0 | 31.8 |

Visual grounding datasets. GoldG+ contains more than 0.8M human-annotated gold grounding data curated by MDETR (Kamath et al., 2021b), including Flickr30K (Plummer et al., 2015), VG Caption (Krishna et al., 2017), GQA (Hudson & Manning, 2019), and RefCOCO (Yu et al., 2016). GoldG removes RefCOCO from GoldG+. Image-text pairs are usually from CC (Sharma et al., 2018) and LocNar (Pont-Tuset et al., 2020).

Table 12 compares our method with MDETR (Kamath et al., 2021b), GLIP (Li et al., 2022b) and X-DETR (Cai et al., 2022) on LVIS minival that contains 5k images. As shown, our detector with RN50x4 as the backbone achieves the leading performance on rare categories without using any annotations of those categories. By contrast, the other three methods employ the costly visual grounding data. MDETR (Kamath et al., 2021b) and X-DETR (Cai et al., 2022) even adopt the ground truth of rare categories but cannot outperform our method. Such results further demonstrate the effectivenss of the proposed SAS-Det.

## B.3 PRESERVING THE KNOWLEDGE FROM THE PRETRAINING

In this section, we explore if models after our finetuning generalizes as well as pretrained models. If so, our finetuning preserves the knowledge learned in the pretraining. Specifically, we evaluated the proposed SAS-Det and *baseline*, which are trained on COCO-OVD with 65 concepts, on LVIS with 1203 concepts. Then, we compare them with the pretrained models and report the results in Table 13. As shown, though finetuned with limited concepts, SAS-Det and *baseline* achieve similar or better performance as pretrained models on rare categories of LVIS. Note that those categories do not appear during finetuning. This indicates that the knowledge learned in the pretraining is successfully preserved in our finetuning. We attribute the improvement on common and frequent categories to the fact that finetuning adapts CLIP to OVD and thus the models learn how to better handle instance-level detection instead of image-level classification that CLIP is pretrained for.

## B.4 ABLATION STUDIES ON LVIS

In this section, we provide ablation studies on LVIS, which are similar as what we did on COCO. Except trained on LVIS-OVD, the baseline model *LVIS baseline* is the same as *baseline* in the main paper. As shown in Table 14, we have consistent observations on LVIS as on COCO. First, based on the results of *LVIS baseline* and *(1)*'s, it is beneficial to remove noisy pseudo boxes from finetuning. Second, *(3)*'s outperforms both *LVIS baseline* and *(2)*'s, which demonstrates that the proposed SAF head helps. The improvement probably comes from the fact that the SAF head avoids noisy pseudo boxes as supervision and incorporates a fusion from different branches.

Table 13: Preserving the knowledge from the pretraining. Models are trained on COCO-OVD and evaluated on the LVIS validation set. Most LVIS categories are unseen during training.

| Method | $AP_r$ | $AP_c$ | $AP_f$ | AP |
|---|---|---|---|---|
| Original CLIP | 8.7 | 6.7 | 4.0 | 6.0 |
| RegionCLIP pretraining | 9.7 | 7.1 | 4.3 | 6.4 |
| *baseline* | 9.2 | 9.7 | 9.0 | 9.4 |
| SAS-Det (Ours) | **13.1** | **10.5** | **9.9** | **10.7** |

Table 14: Ablation studies to analyze the effect of components of SAS-Det on LVIS-OVD.

| Ablation | $AP_r$ | $AP_c$ | $AP_f$ | AP |
|---|---|---|---|---|
| *LVIS baseline* | 19.8 | 25.3 | 31.5 | 26.8 |
| *(1)* Use noisy pseudo boxes to train box regression | (-4.5) 15.3 | 24.0 | 31.1 | 25.2 |
| *(2)* No pseudo labels, train with base data only | (-3.1) 16.7 | 27.0 | 33.0 | 27.6 |
| *(3)* SAF head, fuse the open- and the closed-branches | (+1.1) 20.9 | 26.1 | 31.6 | 27.4 |

## B.5 INFERENCE WITH DIFFERENT RPNs

The proposed SAS-Det leverages an external RPN to get region proposals, and thus it is open to other RPNs without any further finetuning. As shown in Table 15, our model is evaluated together with several RPNs that are trained with different data. Based on those results, we have several findings. First, the model is only trained with *(4)*'s RPN but achieve similar or better performance with other RPNs. This indicates that our model does not rely on the specific RPN that is used in the training, and it is open to different RPNs. Second, *(4)*'s performance is close to others on novel categories, but its RPN is trained without any boxes of novel categories. This demonstrates that the RPN trained on base categories generalizes to novel ones.

## B.6 USING EXTERNAL PLs AT THE BEGINNING

At the beginning of self-training, our PLs are not as good as VL-PLM's Zhao et al. (2022), but our training pipeline allows us to leverage external high quality PLs before the first update to the teacher. To demonstrate this, we trained *baseline* on COCO-OVD using VL-PLM's PLs before the first update. As shown in Table 16, with VL-PLM's PLs, our model improves from 31.4 to 33.5 for novel categories.

## C LIMITATIONS

This work has the following limitations that can be further investigated: (1) Compared to standard training, our self-training involves an additional teacher model, which requires more GPU memory. One possible solution to reduce the cost is to alternatively run the teacher and the student. (2) Although much faster than prior methods, our online pseudo labeling still induces overhead during training when millions of iterations are required. One possible solution is to generate offline PLs each time the teacher model is updated. (3) Although achieving good performance, SAS-Det still suffers from two major failure cases, which are visualized in Sect. D.2. The future work may explore stronger pretrained VLMs and better denoising steps for solutions.

## D QUALITATIVE RESULTS

### D.1 VISUALIZATIONS OF PSEUDO LABELS

Figure 5 provides failure cases of our final PLs (after all updates) on the COCO dataset. We find two major types of failures. (a) Redundant boxes. In this case, one object has multiple predictions that

Table 15: Evaluations with different RPNs on COCO-OVD.

| Training boxes for RPNs | $AP_{50}^{novel}$ | $AP_{50}^{base}$ | $AP_{50}^{all}$ |
|---|---|---|---|
| *(4)* COCO Base (48 categories) | 31.4 | 55.7 | 49.4 |
| *(5)* COCO Base + Novel (65 categories) | 32.7 | 55.7 | 49.7 |
| *(6)* COCO (80 categories) | 32.9 | 55.7 | 49.8 |
| *(7)* LVIS Base (866 categories) | 32.9 | 55.2 | 49.3 |

Table 16: Using PLs of VL-PLM Zhao et al. (2022) in our self-training. Results on COCO-OVD are reported.

| Method | $AP_{50}^{novel}$ | $AP_{50}^{base}$ | $AP_{50}^{all}$ |
|---|---|---|---|
| *baseline* | 31.4 | 55.7 | 49.4 |
| *baseline* + VL-PLM's PLs | 33.5 | 55.9 | 50.1 |

are overlapped with each other. Those overlapped PLs indicate that the pseudo boxes are extremely noisy and cannot be improved by a simple thresholding based on classification confidences. Thus, it is necessary to handle the noise in pseudo boxes separately. We believe that redundant PLs are caused by the poor localization ability of CLIP that provides noisy initial PLs. Though our fine-tuning improve the localization ability to some extent, how to further improve this ability is still challenging and requires future research. (b) Wrong categories. We find the teacher model tends to classify every object into given concepts and generates PLs with wrong categories. Fortunately, there are usually some connections or similarities between the detected objects and the wrong categories. OVD employs text embeddings as classifiers, and text embeddings of related concepts share similarities. For example, the embeddings of "bus" and "train" are close to each other. Thus, though with wrong categories, those PLs may still provide supervision for OVD to some extend.

Figure 6 visualizes more PLs with different times of updates to the teacher model. All samples come from the COCO dataset. As shown, PLs before the update are noisy. Updates remove the noise and improve the quality of PLs.

## D.2 VISUALIZATIONS OF OUR DETECTOR FOR OVD

We visualize good cases and failure cases of the final detector in Fig. 7 and Fig. 8, respectively. All samples come from the COCO dataset, and only predictions of novel categories are provided. As shown in Fig. 7, our detector is able to detect rare objects, e.g. a toy umbrella, a bus with rich textures, and an elephant sculpture.

We find two major failure cases as shown in Fig. 8. (a) Missing instances that usually happen in images with a crowd of objects belonging to one category. In our view, such cases are difficult for fully supervised object detection, let alone OVD. (b) Redundant predictions. We believe this is caused by using redundant PLs (see examples in Fig. 5a) as supervision. Improving PLs will alleviate redundancy in predictions.

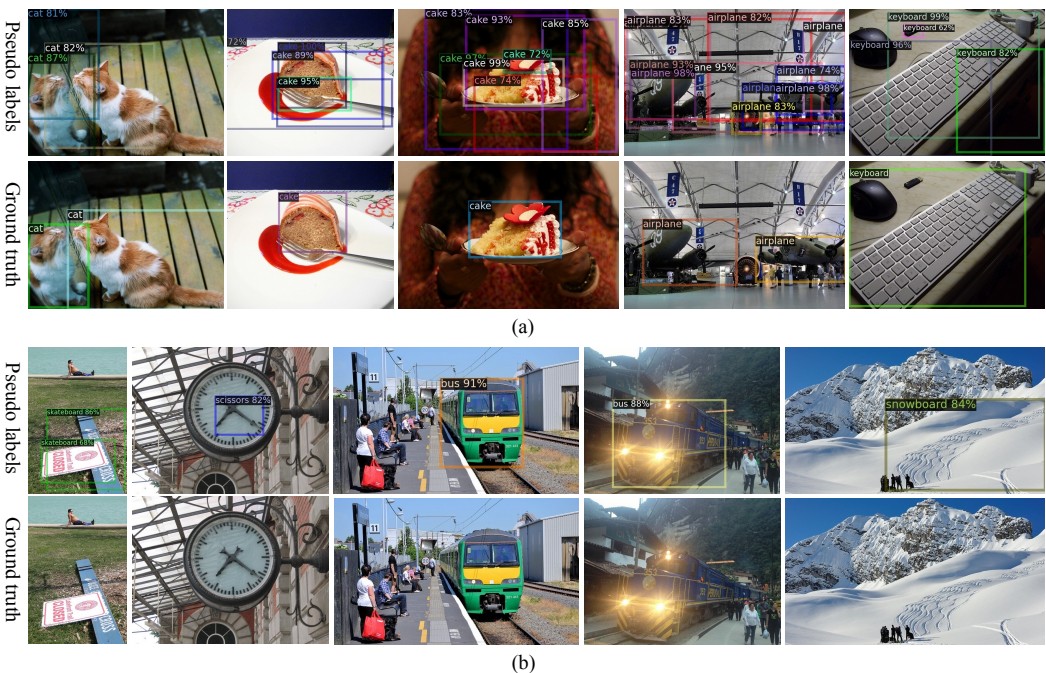

Figure 5: Visualizations of failure cases in PLs after three updates. All samples are from COCO. Two major types of failures: **(a)** Redundant boxes. **(b)** Wrong categories.

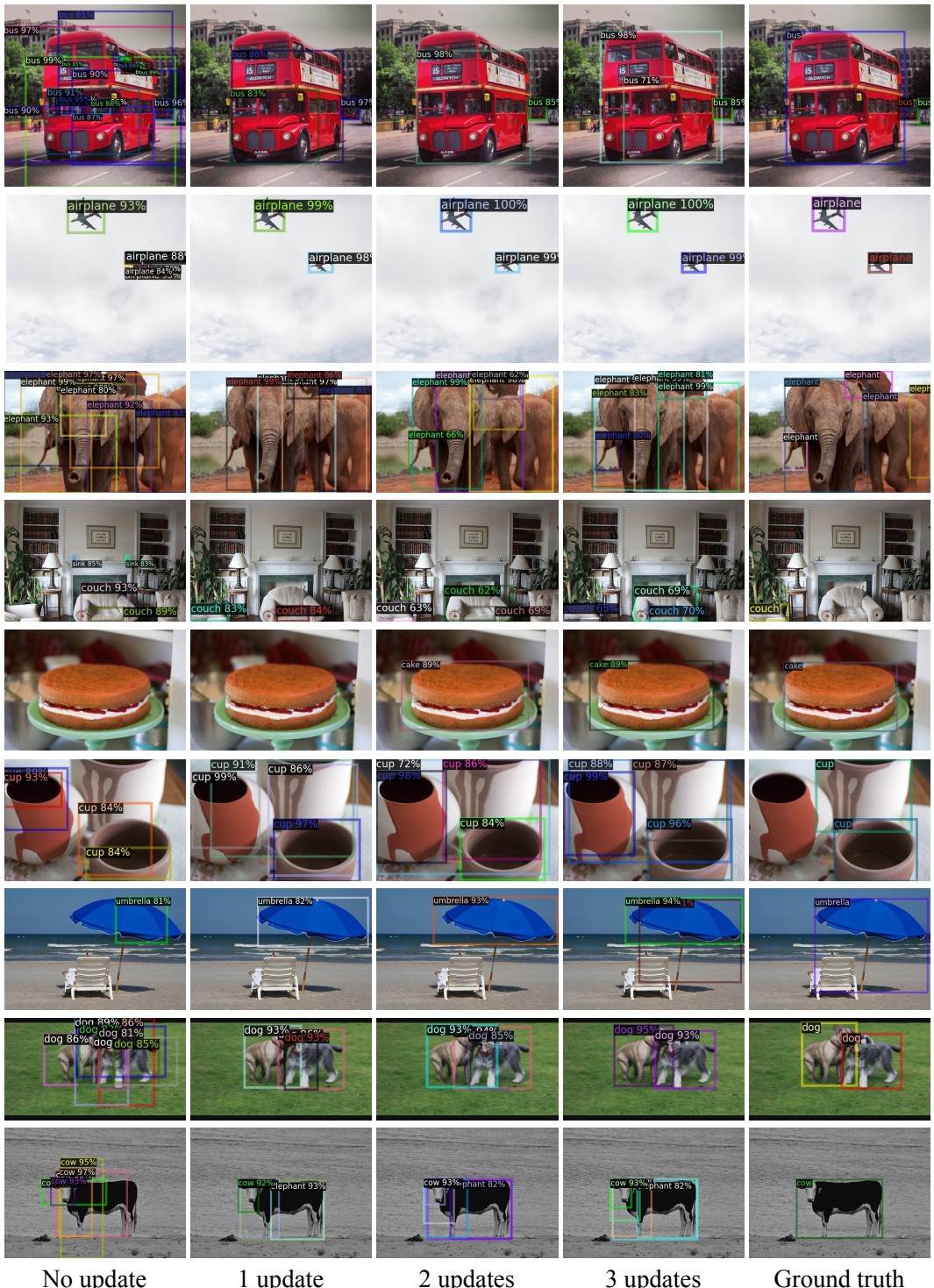

Figure 6: Visualizations of PLs with different numbers of updates for several COCO samples. Updates remove the noise and improve the quality of PLs

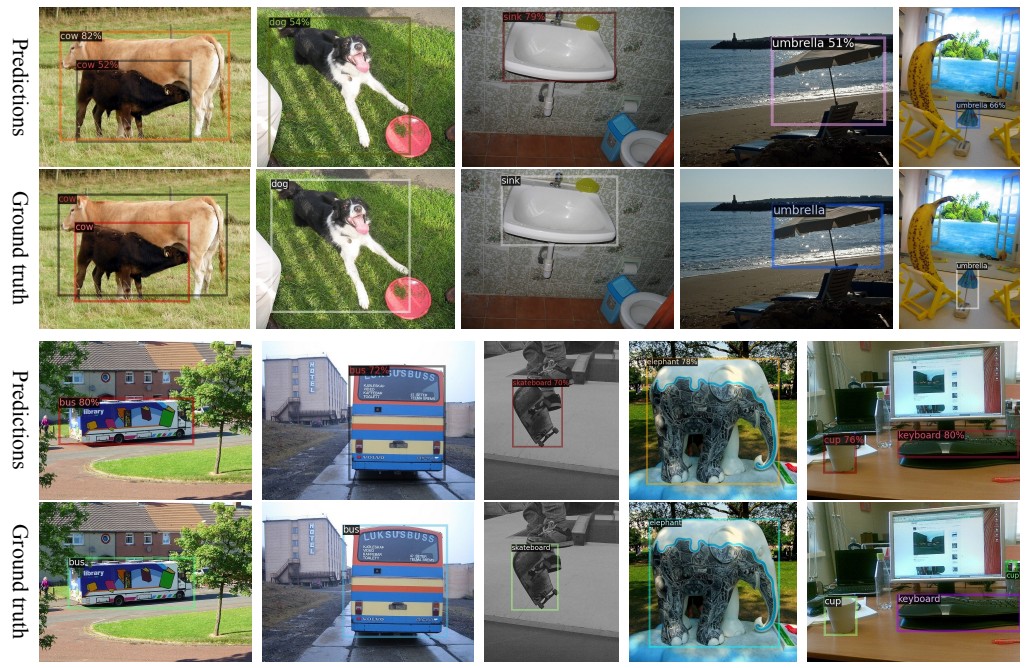

Figure 7: Good cases of the final detector on COCO. Only objects of novel categories are provided. Rare objects can be detected, e.g. a toy umbrella, a bus with rich textures, and an elephant sculpture.

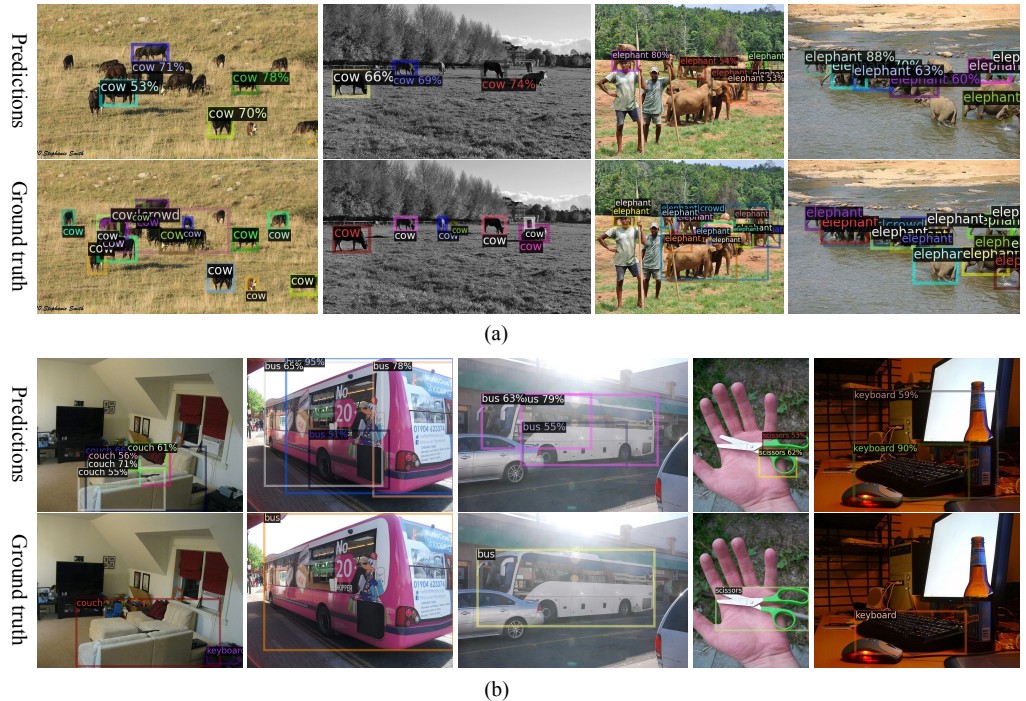

Figure 8: Failure cases of the final detector on COCO. Only objects of novel categories are provided. Two major types: **(a)** Missing instances. **(b)** Redundant predictions.

