# OpenReview forum: "Taming Self-Training for Open-Vocabulary Object Detection"
_ICLR.cc/2024/Conference — ICLR 2024 Conference Withdrawn Submission_

### Official Review · Reviewer_rX3B · 2023-10-30

**Soundness:** 2 fair
**Presentation:** 3 good
**Contribution:** 2 fair
**Rating:** 6
**Confidence:** 3

**Summary:**

This paper address Open-vocabulary object detection (OVOD) by utilizing pseudo labels (PLs) from pretrained vision and language models (VLMs). Authors claim there are two main two challenges of using self-training in OVD: noisy PLs from VLMs and frequent distribution changes of PLs. Two key components: a split-and-fusion (SAF) head and a periodic update strategy are proposed for these challenges. The split-and-fusion (SAF) head splits a standard detection into an open-branch and a closed-branch, and the open-branch only supervised by classification loss without noisy bounding box. The periodic update strategy decrease the number of updates to the teacher, thereby decreasing the frequency of changes in PL distributions, which stabilizes the training process. Experiments show the effectiveness of the proposed modules.

**Strengths:**

1.	The paper is well-written and easy following, the writing is good.
2.	The two problems mentioned in this paper, i.e., reducing noisy PLs and make updating of teach model stable, are very important in self-training.
3.	The experiment on two benchmarks achieves good performance.

**Weaknesses:**

1.	Although two problems found in this paper are key, but the method proposed for addressing the problems are trivial, especially, the periodic update, periodically updating the teacher after a set number of iterations. It’s just like a trick.
2.	Missing the theoretical and experimental analysis for the updating of teaching model and the proposed periodic update.
3.	The architecture of the teach model is not clear.
4.	It’s better to see the experiments that only train the open branch with novel classes, which would help to understand the effect of the open-branch.

**Questions:**

See weakness.

---

### Official Review · Reviewer_NEiz · 2023-10-31

**Soundness:** 2 fair
**Presentation:** 3 good
**Contribution:** 3 good
**Rating:** 5
**Confidence:** 5

**Summary:**

This paper proposes to use self-training for open-vocabulary object detection. Based on an external and frozen region proposal network, the paper change online update to periodic update, and use split-and-fusion head for separately handling open and close categories and fuse their scores in prediction.

**Strengths:**

The paper makes self-training work for open-vocabulary object detection and obtains state-of-the-art results, which are non-trivial given prior attempts that are not so simple. Most of the modifications are verified by experiments, thus providing insights to the community for future research.

**Weaknesses:**

1. The hypothesis claimed in this paper is not verified. Most modifications are motivated by the idea that pseudo-labels are noisy and change frequently. However, it is not confirmed by any analysis. The periodic update strategy and the SAF head are indeed effective, but it does not mean that they work because they reduce the noise in pseudo labels.
2. The ablation study does not reveal the entangled effects between components. For example, what if the baseline uses SAF head but does not use external RPN ((6) in Table 3 with (1))?
3. The external region proposal network seems to be an empirical modification that is tried during experiments without specific motivations and analysis. This appears to be vital to make the framework (brings 6.0 AP of novel categories) but makes it unfair and not so elegant in comparison with previous works. Can the paper provide more analysis about it?

**Questions:**

See weakness.
Is there any prior study or evidence showing that pseudo-labels are noisy and unstable during training?

---

### Official Review · Reviewer_vFm8 · 2023-11-02

**Soundness:** 3 good
**Presentation:** 3 good
**Contribution:** 3 good
**Rating:** 5
**Confidence:** 4

**Summary:**

This paper studies the self-training problem in open-vocabulary object detection (OVD). Two challenges are identified in this work, including noisy pseudo labels and frequent distribution changes of pseudo labels during self-training. To solve these two challenges, this paper proposes a split-and-fusion (SAF) head for self-training in the framework of OVD, and a periodic update strategy for the teacher model in the framework of teacher-student self-training framework. Extensive experiments are presented to validate the effectiveness.

**Strengths:**

- This paper is well-written and well organized.
- Extensive experiments are conducted to validate the effectiveness.

**Weaknesses:**

- I acknowledge the importance to study the self-training problem in open-vocabulary object detection. My main concern is the technique novelty about the split-and-fusion head for self-training and the periodic update strategy for the teacher model. As far as I know, the proposed split-and-fusion head is quite similar to the noise-bypass head in [1] for semi-supervised object detection, and the proposed periodic update strategy is quite similar to the periodic update strategy in [2] for source-free domain adaptive object detection. Both semi-supervised object detection and source-free domain adaptive object detection belong to the field of self-training. This work didnot provide an in-depth discussion about the relation between the proposed methods and the existing self-training methods. So can you explain the difference between your methods and the listed methods [1,2], although the task in your work is open-vocabulary object detection, which is different from [1,2].

[1] Dual Decoupling Training for Semi-Supervised Object Detection with Noise-Bypass Head. AAAI2022.

[2] Periodically Exchange Teacher-Student for Source-Free Object Detection. ICCV 2023.

**Questions:**

I will reconsider the rating after reading the rebuttal and discussing with other reviewers.

---

### Official Review · Reviewer_wquL · 2023-11-05

**Soundness:** 1 poor
**Presentation:** 1 poor
**Contribution:** 1 poor
**Rating:** 3
**Confidence:** 4

**Summary:**

This paper proposes a teacher-student self-training based open vocabulary object detection framework, SAS-Det, which addresses the self-training difficulties of OVOD from two aspects.

Firstly, this work designs head branches for close-set and open-set respectively, claiming to solve the noise introduced by pseudo-labels.

Secondly, it proposes a staged update strategy for teacher-student self-training.

Based on these two methods, the proposed framework achieves promising performance on the COCO & LVIS open vocabulary benchmarks.

**Strengths:**

* The intention of this paper is clear, focusing on how to achieve effective self-training in Open Vocabulary Object Detection (OVOD) and achieving promising results on the COCO and LVIS datasets.

**Weaknesses:**

Overall, this paper addresses the challenging problem of open vocabulary from the perspective of pseudo label. I believe the main issues with this paper are as follows:
* Teacher-student self-training is a classic framework in semi-supervised object detection. However, the proposed periodic update strategy in this paper actually only modifies the frequency of teacher network updates, which lacks any innovation. In addition, the author claims that reducing the frequency of teacher network updates can decrease the frequency of changes in PL distributions, thereby stabilizing the training process. This is curious because this issue also exists in semi-supervised object detection, yet it does not have a similar effect in the ssod field. There is a lack of quantitative analysis here, and I believe this design deviates from the theme of open vocabulary object detection.

* The overall framework appears particularly complex, making the paper difficult to understand.

* In Figure 2b, open concepts enter the clip encoder through prompts, but how are these prompts designed? The paper does not provide a detailed explanation.

* The paper details the use of an external RPN in the method section, which yields better results than F-VLM. However, the paper does not provide a detailed explanation of how the specific RPN is trained, whether it uses additional training data, and whether the RPN is within the main contributions of this paper. However, the paper spends a considerable amount of space introducing external RPN, which is confusing.

Based on the aforementioned reasons, I believe that this paper requires improvement in terms of writing, experimentation, and organization to be accepted by ICLR. The current quality of the paper clearly does not meet the acceptance level of ICLR.

**Questions:**

please refer to weakness